# Gallbladder Polyp Classification in Ultrasound Images Using an Ensemble Convolutional Neural Network Model

**DOI:** 10.3390/jcm10163585

**Published:** 2021-08-14

**Authors:** Taewan Kim, Young Hoon Choi, Jin Ho Choi, Sang Hyub Lee, Seungchul Lee, In Seok Lee

**Affiliations:** 1Department of Mechanical Engineering, Pohang University of Science and Technology, Pohang 37673, Korea; twkim97@postech.ac.kr; 2Department of Internal Medicine, College of Medicine, The Catholic University of Korea, Seoul 06591, Korea; crzyzs@naver.com; 3Department of Internal Medicine and Liver Research Institute, Seoul National University Hospital, Seoul National University College of Medicine, Seoul 03080, Korea; pseudo.jh@gmail.com (J.H.C.); gidoctor@snu.ac.kr (S.H.L.); 4Graduate School of Artificial Intelligence, Pohang University of Science and Technology (POSTECH), Pohang 37673, Korea; 5Institute of Convergence Research and Education in Advanced Technology, Yonsei University, Seoul 03722, Korea

**Keywords:** gallbladder polyp, ultrasonography, deep learning, true polyp, differential diagnosis

## Abstract

Differential diagnosis of true gallbladder polyps remains a challenging task. This study aimed to differentiate true polyps in ultrasound images using deep learning, especially gallbladder polyps less than 20 mm in size, where clinical distinction is necessary. A total of 501 patients with gallbladder polyp pathology confirmed through cholecystectomy were enrolled from two tertiary hospitals. Abdominal ultrasound images of gallbladder polyps from these patients were analyzed using an ensemble model combining three convolutional neural network (CNN) models and a 5-fold cross-validation. True polyp diagnosis with the ensemble model that learned only using ultrasonography images achieved an area under receiver operating characteristic curve (AUC) of 0.8960 and accuracy of 83.63%. After adding patient age and polyp size information, the diagnostic performance of the ensemble model improved, with a high specificity of 88.35%, AUC of 0.9082, and accuracy of 87.61%, outperforming the individual CNN models constituting the ensemble model. In the subgroup analysis, the ensemble model showed the best performance with AUC of 0.9131 for polyps larger than 10 mm. Our proposed ensemble model that combines three CNN models classifies gallbladder polyps of less than 20 mm in ultrasonography images with high accuracy and can be useful for avoiding unnecessary cholecystectomy with high specificity.

## 1. Introduction

Gallbladder (GB) polyps are tissue growths that protrude from the GB wall into the lumen: they can be classified into pseudopolyps, represented by cholesterol polyps, and true polyps, including adenoma and adenocarcinoma [1,2]. The most commonly used imaging modality for the diagnosis and follow-up of GB polyps is abdominal ultrasound [3]. GB polyps are known to be found in approximately 5% of the patients who undergo abdominal ultrasound, and the number of cases of GB polyps being incidentally detected is increasing with the recent increase in abdominal ultrasound examinations that are being conducted as part of regular health check-ups [4]. True GB polyps are already malignant or have malignant potential; therefore, cholecystectomy is required. However, it is difficult to differentiate between true and pseudo polyps through pre-operative examinations, including abdominal ultrasound [3,5].

Several studies have investigated the risk factors of neoplastic (true) polyps by combining ultrasound findings and clinical factors [6,7,8]. One of the well-known risk factors of neoplastic GB polyps is the polyp size; typically, 10 mm is used as the cutoff value for cholecystectomy [9]. However, Wennmacker et al. reported that, with a cutoff value of 10 mm, the sensitivity and specificity for diagnosing neoplastic polyps are approximately 70% and that approximately 30% of the patients undergo unnecessary cholecystectomy [10]. It has been reported that GB polyps in approximately 94% of patients are diagnosed as malignant GB polyps if the polyp size is more than 20 mm [11]. Therefore, it is a good strategy to consider cholecystectomy first if the size of the GB polyp is greater than 20 mm. In contrast, polyps of less than 20 mm require more accurate differential diagnosis.

Morphological features of polyps in abdominal ultrasound, such as hyperechoic spots and sessile shape, aid in the differential diagnosis of neoplastic GB polyps [8]. However, differential diagnosis using such imaging features of GB polyps is ineffective because accurate evaluation of these features is difficult, and results may vary depending on the ultrasound image reader. Recent advancements in deep learning have yielded impressive results in various fields, especially in medical imaging, where deep learning has been used to effectively distinguish imperceptible differences in image patterns [12]. Several studies have used deep learning to analyze medical images, but only some have focused on GB polyps [12,13]. Therefore, in this study, we aimed to differentiate true GB polyps using deep learning, especially those less than 20 mm, which require clinical differential diagnosis.

## 2. Materials and Methods

### 2.1. Patients and Dataset Preparation

This study was conducted on 529 patients with GB polyps in Seoul St. Mary’s Hospital and Seoul National University Hospital. All patients were diagnosed with GB polyps through abdominal ultrasound and underwent cholecystectomy at one of the hospitals between January 2008 and February 2020. We excluded patients with strongly suspected GB cancer based on pre-operative computed tomography or magnetic resonance imaging (*n* = 9) and those with polyp sizes greater than 20 mm (*n* = 19); consequently, a total of 501 patients were enrolled in this study.

We established a gold standard for diagnosing GB polyps for training artificial intelligence models and evaluating the diagnostic performance of artificial intelligence models. This gold standard was the pathological diagnosis of GB polyps obtained from the cholecystectomy pathology report. Ultrasound images were obtained using ultrasound scanners (Siemens Acuson S2000, Siemens Healthcare, Erlangen, Germany or GE Healthcare LOGIQ E9 or E10, GE healthcare, WI, USA) by radiologists and gastroenterologists with about 5 to 20 years of experience in transabdominal ultrasonography. Ultrasound images with GB polyps were selected, and the collected ultrasonography images were preprocessed for the analysis as follows. First, we cropped the polyps from the ultrasound images and rescaled all the cropped images to the same size. Sample images are shown in Figure 1. We normalized the brightness and contrast of the images to minimize the effects of different devices. A 5-fold cross-validation was performed to ensure that the results were unaffected by the separation of the dataset into the training and validation sets. The entire dataset was randomly divided into five equal sub-datasets. Four of the five sub-datasets were used for training, and one was used for validation; we changed the validation set each time. To achieve a superior classification performance, we conducted 8-fold augmentation on the images in the training dataset by rotating the images by 90° three times and flipping them horizontally for each rotation. The deep learning model was trained with these prepared datasets, and 5-fold results were used to perform the ultrasonography image classification and patient-specific diagnoses. A flow chart of the study procedure is shown in Figure 2.

This study was conducted in accordance with the Declaration of Helsinki. The study protocol was approved by the Institutional Review Boards (IRB) of each institution (Seoul St. Mary’s Hospital IRB No. KC20RIDI0989, Seoul National University Hospital IRB No. 2002-097-110), and informed consent was waived due to the retrospective nature of the study.

### 2.2. Ensemble Convolutional Neural Network Model

We propose a new ensemble model for GB polyp classification. This ensemble model combines three convolutional neural network (CNN) models that are widely used in computer vision for image recognition (ResNet [14], Inception v3 [15], and DenseNet [16]). Each model operates independently and extracts features utilizing its own method. ResNet extracts deep hierarchical features using skip connection, Inception v3 extracts multi-scale features using parallel convolution layers with different kernel sizes, and DenseNet extracts dense features through the direct connection of features from shallow to deeper layers. The ensemble model was constructed by connecting the aforementioned models in parallel and summing the results of each model.

The process followed by the ensemble model to perform prediction is as follows: an ultrasonography image is input to the input layers of individual models; each model predicts the probability for the input; and, finally, the probabilities of each model are summed to calculate the final prediction probability for the input image. The probability for an image calculated through the above process is summed up for each patient to diagnose whether that patient’s polyp is true or pseudo. The schematic of the ensemble model is illustrated in Figure 3.

### 2.3. Clinical Validation

We first compared the results of each CNN model with those of the ensemble CNN model to confirm its effectiveness. Thereafter, we compared the classification performance when patient age and polyp size, which are known risk factors of neoplastic (true) GB polyps, were selectively added. The comparison of diagnostic performance according to the additional clinical information was conducted in three ways: with only polyp size added, only patient age added, and both polyp size and patient age added. Additional clinical information was concatenated parallel to the last fully connected layer of each model, and this clinical information was used along with the ultrasonography features for the final prediction. We compared the classification performance of the ensemble model by dividing the data into two groups with polyp sizes larger and smaller than 10 mm. Model training was performed under the same conditions and with the same hyper-parameters, learning rate, batch size, and number of epochs.

### 2.4. Performance Measures and Statistics

The final diagnosis of the ensemble model was determined depending on whether the aggregated probability value exceeds a threshold. We set the thresholds for each fold such that they maximized the Youden index, where the sum of sensitivity and specificity was the highest, and the classification was performed according to these thresholds. For the evaluation of the GB polyp classification results, we used the performance measures of sensitivity, specificity, positive predictive value (PPV), negative predictive value (NPV), accuracy, and area under the receiver operating characteristic (ROC) curve (AUC). Diagnostic performance measures were calculated using the mean and 95% confidence interval of the 5-fold cross-validation results.

Student’s *t*-test was conducted on age and polyp size between the pseudo and true polyp groups to verify whether they are statistically significantly different. Cutoff values were set for the age and size of polyps to observe the effects of these values in differentiating between pseudo and true polyps. These cutoff values were calculated using ROC analysis and the Youden index, and the sensitivity, specificity, and accuracy were calculated according to the calculated values. A *p*-value of less than 0.05 was considered statistically significant. The statistical analyses were performed using Python version 3.6.8 (Wilmington, DE, USA), SciPy 1.1.0 (Austin, TX, USA), and Numpy version 1.19.4 (Austin, TX, USA).

## 3. Results

### 3.1. Dataset Composition and Characteristics

This study included a total of 501 patients, out of which 412 had pseudopolyps and 89 had true polyps. The images used for the deep learning analysis contained 1039 pseudopolyps and 421 true polyps. The mean ages and polyp sizes were significantly higher in the true polyp group compared to the pseudopolyp group (59.1 vs. 48.3 years and 12.6 vs. 10.5 mm, respectively). The specifications of the dataset are summarized in Table 1.

The cutoff values for age and polyp size were 52 years and 13 mm, respectively. With a polyp size cutoff value of 13 mm, the accuracy was 76.65%, sensitivity was 42.70%, and specificity was 83.98%. With a patient age cutoff value of 52 years, accuracy was 62.28%, sensitivity was 77.53%, and specificity was 58.98%. These results show that the pseudo and true polyp groups were not sufficiently divided by polyp size and patient age. The distributions of the pseudopolyp and true polyp groups’ age and polyp size are shown in Figure 4.

### 3.2. Diagnostic Performance of the Ensemble Model

The classification results of the individual models and ensemble model are listed in Table 2. When training with ultrasonography without clinical information, the individual ResNet152, Inception v3, and DenseNet161 models achieved AUCs of 0.8710, 0.8625, and 0.8776 and accuracies of 80.39%, 81.76%, and 83.84%, respectively, and the ensemble model achieved an AUC of 0.8960 and accuracy of 83.63%. When training with ultrasonography and the clinical information, the ensemble model achieved an AUC of 0.9082, accuracy of 87.61%, sensitivity of 84.28%, specificity of 88.35%, PPV of 62.42%, and NPV of 96.31%; these values are higher than those of the individual models and higher than those when age and polyp size were not considered. The ROC curves according to the addition of clinical information are shown in Figure 5.

### 3.3. Diagnostic Performance Based on the GB Polyp Size

We compared the performance of the ensemble model with both age and polyp size information according to the polyp size. The results are listed in Table 3. For the case of 370 patients with polyp sizes of 10 mm or more (pseudopolyp: 299, true polyp: 71), the ensemble model achieved an accuracy of 87.15%, sensitivity of 85.30%, specificity of 87.64%, and AUC of 0.9131 with a threshold of 0.287. For the case of 131 patients with polyp sizes less than 10 mm (pseudopolyp: 113, true polyp: 18), the ensemble model achieved an accuracy of 86.61%, sensitivity of 93.33%, specificity of 85.57, and AUC score of 0.8942 with a threshold of 0.292.

## 4. Discussion

We proposed a new ensemble model that combines three CNN models (ResNet, Inception v3, and DenseNet) that have shown excellent performance in the field of image recognition [17]. Each model has different feature extraction characteristics, and the ensemble model combines these features to obtain the final result. The model utilizes more diverse morphological and texture features and extracts more informative features useful for GB polyp classification than when using only a single model. This structural uniqueness of the ensemble model made it possible to classify GB polyps of less than 20 mm from abdominal ultrasound images with high accuracy. We improved the diagnostic performance of this model by adding information such as age and polyp size, known as the risk factors for neoplastic (true) GB polyps, to the ensemble model.

To date, few studies have analyzed abdominal ultrasound images of GB polyps using artificial intelligence. Yuan et al. performed computer-assisted image analysis using spatial and morphological features from ultrasound images of GB polyps and achieved an accuracy of 87.5% in neoplastic polyp diagnosis [18]. Although their study used computer-aided analysis, it differed from ours in that it did not use a CNN. Further, fewer than 100 subjects were enrolled in their study. Jeong et al. reported the classification of GB polyps using a CNN and achieved an accuracy of approximately 85.7% [19]. In their study, the maximum GB polyp size was approximately 47 mm, mean polyp size of the neoplastic polyp group was 18.4 mm, and it is highly likely that a large number of polyps with a size of 20 mm or more were included. Considering that most polyps with a size of 20 mm or more in previous studies were found to be neoplastic polyps, their study likely included several polyps that were relatively easy to differentiate clinically based on size alone, which may have resulted in an overestimation of the diagnostic accuracy [11,20]. Our study excluded polyps larger than 20 mm to only target patients with GB polyps that are difficult to discriminate in real clinical practice. Nevertheless, our model showed similar diagnostic performance and numerically better accuracy compared to the model by Jeong et al. This is likely because our ensemble model employs two additional CNNs in addition to Inception v3, the only CNN used in their study. Interestingly, in the present study, when age and size factors were not considered, our model showed a diagnostic performance similar to that of the single DenseNet model. This is because our model uses the results of the two other models in addition to those of the DenseNet; therefore, if the performance of the two other models is insufficient, the performance of the ensemble model will be similar to that of the DenseNet model, which has the best performance of all single models. Meanwhile, when age and size factors were added, each of the three models achieved a certain level of performance and complementary effect, and the performance improvement effect of the ensemble model was maximized compared to the single models, thereby yielding the best performance.

Currently, one of the most important factors for differentiating neoplastic GB polyps in clinical practice is the polyp size, and a recent guideline suggested a cutoff value of 10 mm [9]. However, this value is not sufficient to differentiate neoplastic polyps. According to a recent study, when cholecystectomy was performed with a 10 mm criterion, approximately 40% of the patients had non-neoplastic (pseudo) polyps and did not require cholecystectomy [21]. In the present study, the diagnostic accuracy of the model, when the size factor was not included, was approximately 84%, indicating that the size factor is not absolutely crucial. Nevertheless, this suggested cutoff value is widely applied in clinical practice, and as such, we analyzed the performance of our model in each of the subgroups based on this 10 mm threshold. As a result, the diagnostic performance for true polyps was best in the subgroup with a polyp size greater than 10 mm and an AUC of 0.9131. This is a positive factor in our model, considering that in real clinical practice, it is often necessary to differentiate true polyps from polyps greater than 10 mm rather than those that are smaller than 10 mm. Furthermore, our model showed a high specificity of nearly 90% and a PPV of approximately 62%, which is considerably higher than the 14.9% PPV for the diagnosis of neoplastic GB polyps of abdominal ultrasound reported in a recent systematic review [5]. These high specificity and PPV values of our model can significantly reduce false positive diagnoses for neoplastic (true) polyps, avoiding unnecessary cholecystectomy. In fact, when we evaluated the ultrasound images selected by our model, we found that our model could effectively discriminate large, sessile pseudopolyps that could be mistaken for true polyps. Our model could effectively distinguish true polyps among the small polyps with hyperechoic spots, which are generally considered pseudopolyps. Figure 6 shows the true and false cases of the model prediction for representative ultrasound images of pseudo and true polyps based on the polyp size.

Our study has several strengths. First, we developed a new ensemble model that combines three CNNs. To the best of our knowledge, no studies have demonstrated a model that incorporates multiple CNN models for the differential diagnosis of GB polyps. Second, we limited the study subjects to include only GB polyps that are difficult to distinguish in actual clinical practice and enrolled the largest number of patients among the studies with this subject limitation.

There are some limitations to this study. First, the model was trained and validated only on still images, and it could not include all the information of ultrasound images observed in real-time videos, as in an actual test. To further improve the model and make it more practically applicable using ultrasound video information, it is necessary to develop an algorithm that distinguishes the features of polyps in ultrasound videos; we are planning a follow-up study in this direction. Second, to exclude patients whose pathological diagnoses of polyps are not confirmed, only those who had undergone cholecystectomy were included in this study, and a selection bias may have occurred as a result of this. Future prospective studies including patients who have not undergone cholecystectomy are required to further verify our ensemble model. Third, there is a limit to the resolution of transabdominal ultrasound itself. One method to overcome this is to use an endoscopic ultrasound (EUS), which is known to be more helpful in the differential diagnosis of GB polyps [22]. Thus, we are planning an artificial intelligence analysis study using EUS. In this future study, verifying whether the diagnostic accuracy is good enough to overcome the disadvantages of EUS is necessary, as it is more invasive and less accessible than transabdominal ultrasound.

## 5. Conclusions

We developed a new ensemble model that combines three CNN models and distinguishes true polyps with high accuracy from ultrasound images of patients with GB polyps that are less than 20 mm in size. Our ensemble model, which has relatively high specificity and PPV, can help avoid unnecessary cholecystectomy. Future studies using EUS and real-time ultrasound video are necessary to develop a model with better diagnostic accuracy.

## Figures and Tables

**Figure 1 jcm-10-03585-f001:**
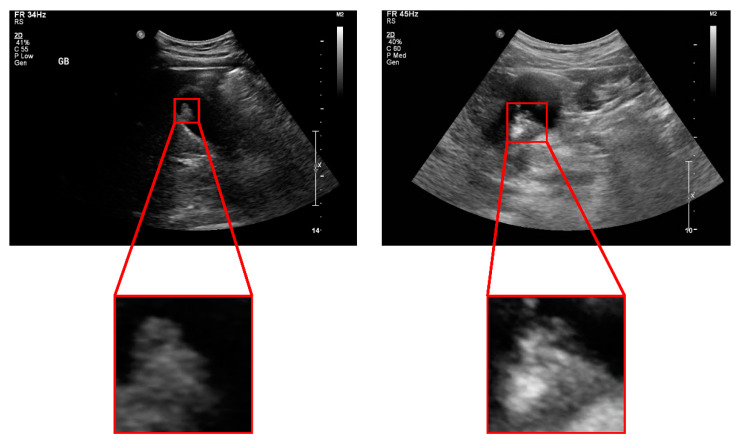
Examples of cropped images. We manually cropped the polyp area from each gallbladder ultrasonography image with a one-to-one ratio. All cropped images were resized to the same size for deep learning analysis.

**Figure 2 jcm-10-03585-f002:**
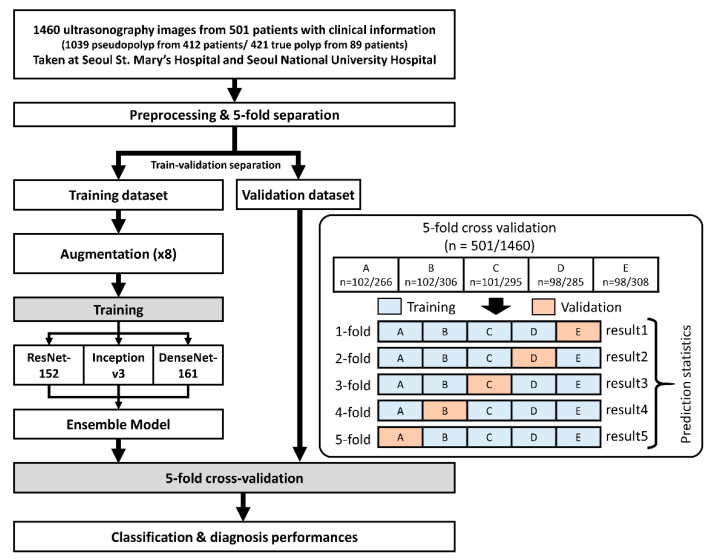
Flow chart of the study procedure. The data collected from our institutions were used for training and validation. A total of 1460 images from 501 patients were randomly separated into five sub-datasets (i.e., A: 102/266, B: 102/306, C: 101/295, D: 98/285, and E: 98/308), and each sub-dataset was used sequentially for training and validation. The sub-dataset results were aggregated to obtain prediction statistics of the model, and patient-specific diagnoses were conducted based on the sub-dataset results where the patients belong.

**Figure 3 jcm-10-03585-f003:**
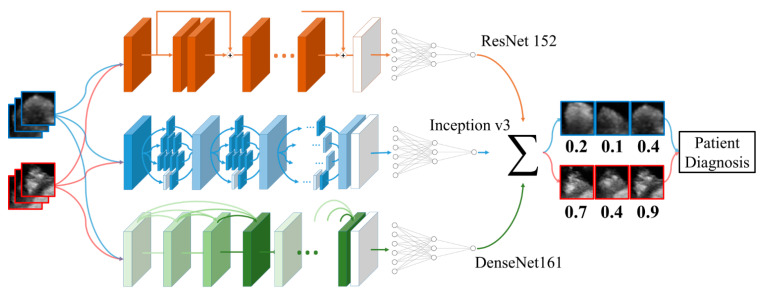
Schematic of the ensemble model of ResNet 152, Inception v3, and DenseNet 161. The transfer learning method was used for the initial parameter setting, and minor fine-tuning was conducted for gallbladder polyp classification. Clinical information was concatenated to the end of each model’s fully connected layer.

**Figure 4 jcm-10-03585-f004:**
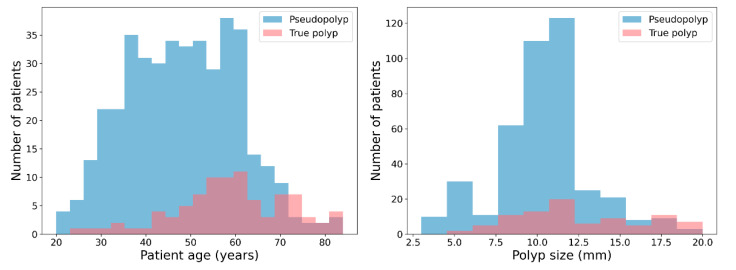
Histograms of patients with pseudopolyp and true polyp by age and polyp size. The distributions of pseudopolyps and true polyps are illustrated by blue and red bars, respectively.

**Figure 5 jcm-10-03585-f005:**
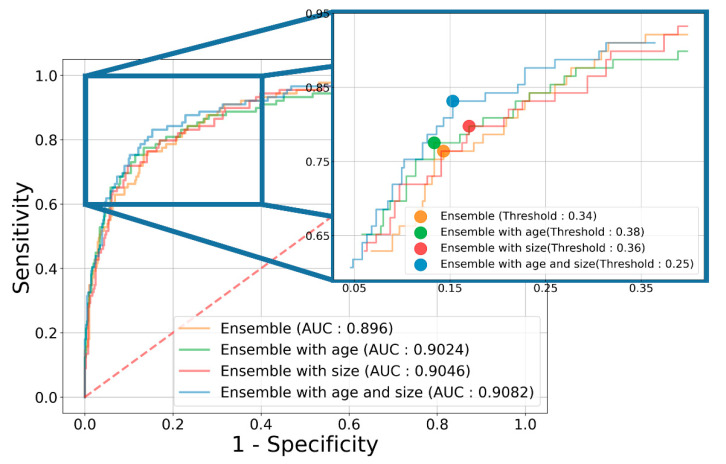
ROC curves of ensemble model according to the addition of clinical information. The orange, green, red, and blue lines show the results from the ensemble model without clinical information, with patient age, with polyp size, and with both, respectively.

**Figure 6 jcm-10-03585-f006:**
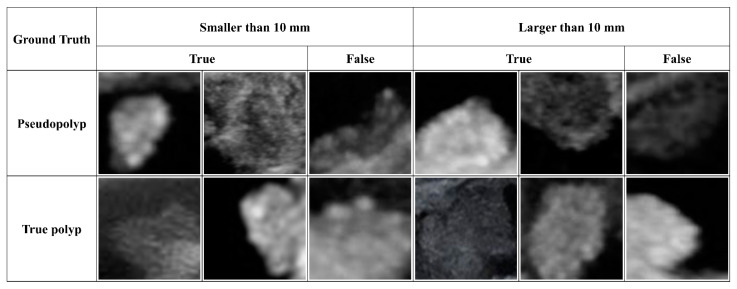
Model prediction results and corresponding images according to GB polyp size. The first and second rows are ultrasound images of pseudopolyps and true polyps confirmed through cholecystectomy, respectively.

**Table 1 jcm-10-03585-t001:** Specifications of study dataset. We compared the pseudopolyp and true polyp groups using Student’s *t*-test.

	Pseudopolyp	True Polyp	*p*-Value
Number of patients	412	89	
Number of images	1039	421	
Age (years)	48.3 ± 12.3	59.1 ± 12.6	<0.001
Polyp size (mm)	10.5 ± 2.8	12.6 ± 3.8	<0.001

**Table 2 jcm-10-03585-t002:** Diagnostic performances of the individual models and ensemble model according to the addition of clinical information. All results were calculated based on the thresholds at which the Youden index was highest. Data in parentheses are 95% confidence intervals. PPV: positive predictive value, **NPV,** negative predictive value.

Clinical Information	Model	Accuracy	Sensitivity	Specificity	PPV	NPV	AUC
Patient Diagnosis							
None	ResNet 152	80.39 (74.48~86.30)	84.28 (74.64~93.92)	79.59 (71.41~87.78)	48.37 (35.94~60.80)	96.00 (93.81~98.19)	0.8710 (0.8335~0.9084)
	Inception v3	81.76 (65.42~98.09)	84.47 (72.67~96.28)	81.29 (59.47~100.0)	56.84 (33.48~80.21)	96.37 (93.78~98.95)	0.8625 (0.7991~0.9260)
	DenseNet 161	83.84 (77.07~90.62)	81.78( 72.36~91.20)	84.23 (74.71~93.76)	54.89 (41.86~67.91)	95.67 (93.91~97.44)	0.8776 (0.8449~0.9103)
	Ensemble	83.63 (77.34~89.93)	84.08 (74.58~93.58)	83.49 (74.61~92.37)	54.48 (40.97~68.00)	96.18 (94.27~98.09)	0.8960 (0.8599~0.9321)
Age	ResNet 152	80.35 (72.55~88.14)	84.47 (74.26~94.69)	79.57 (68.59~90.54)	49.14 (34.99~63.28)	96.07 (93.76~98.38)	0.8701 (0.8394~0.9008)
	Inception v3	81.97 (73.19~90.76)	88.22 (77.70~98.75)	80.81 (68.51~93.10)	52.05 (38.02~66.07)	97.02 (94.34~99.69)	0.8761 (0.8314~0.9208)
	DenseNet161	77.91 (67.64~88.18)	92.89 (83.10~100.0)	74.56 (61.12~88.00)	45.75 (34.25~57.24)	98.20 (95.86~100.0)	0.8825 (0.8330~0.9320)
	Ensemble	84.99 (73.37~96.61)	86.38 (75.30~97.46)	84.70 (69.35~100.0)	60.81 (37.52~84.10)	96.83 (94.70~98.95)	0.9024 (0.8495~0.9554)
Size	ResNet152	79.01 (70.61~87.42)	90.59 (79.58~100.0)	76.45 (65.78~87.12)	46.78 (34.56~59.00)	97.61 (94.73~100.0)	0.8848 (0.8303~0.9393)
	Inception v3	83.26 (78.72~87.80)	81.97 (70.87~93.08)	83.49 (76.16~90.82)	52.77 (45.54~60.00)	95.68 (93.12~98.23)	0.8779 (0.8496~0.9061)
	DenseNet 161	78.40 (66.12~90.67)	89.93 (80.46~99.41)	75.97 (59.57~92.36)	47.58 (33.76~61.40)	97.45 (95.27~99.64)	0.8736 (0.8442~0.9030)
	Ensemble	81.20 (69.71~92.69)	92.04 (87.89~96.19)	78.89 (64.70~93.09)	51.34 (33.94~68.74)	97.91 (97.12~98.71)	0.9046 (0.8537~0.9555)
Age + Size	ResNet 152	79.63 (70.94~88.33)	88.88 (84.57~93.19)	77.68 (66.75~88.60)	47.81 (36.06~59.56)	97.01 (95.86~98.16)	0.8814 (0.8432~0.9196)
	Inception v3	81.63 (70.86~92.40)	82.63 (68.58~96.68)	81.34 (66.28~96.40)	52.99 (33.97~72.01)	95.94 (92.92~98.96)	0.8756 (0.8358~0.9153)
	DenseNet 161	84.63 (81.01~88.25)	85.33 (77.68~92.98)	84.47 (79.94~89.00)	54.64 (46.76~62.52)	96.43 (94.56~98.30)	0.8991 (0.8602~0.9380)
	Ensemble	87.61 (81.03~94.18)	84.28 (72.79~95.76)	88.35 (81.24~95.46)	62.42 (45.42~79.43)	96.31 (93.70~98.92)	0.9082 (0.8550~0.9614)

**Table 3 jcm-10-03585-t003:** Diagnostic performances of the ensemble model according to the polyp size.

Model (With Clinical Info)	Size	Accuracy	Sensitivity	Specificity	PPV	NPV	AUC
Patient Diagnosis							
Ensemble (age + size)	-	87.61 (81.03~94.18)	84.28 (72.79~95.76)	88.35 (81.24~95.46)	62.42 (45.42~79.43)	96.31 (93.70~98.92)	0.9082 (0.8550~0.9614)
	≥10 mm	87.15 (80.62~93.69)	85.30 (70.69~99.91)	87.64 (80.65~94.64)	63.46 (50.20~76.72)	96.13 (92.25~100.0)	0.9131 (0.8523~0.9740)
	<10 mm	86.61 (67.88~100.0)	93.33 (74.82~100.0)	85.57 (64.33~100.0)	59.24 (29.11~89.37)	99.26 (97.20~100.0)	0.8942 (0.7867~1.000)

## Data Availability

The data presented in this study are available on request from the corresponding author. The data are not publicly available due to privacy restrictions.

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
