# Peer review of "Gallbladder Polyp Classification in Ultrasound Images Using an Ensemble Convolutional Neural Network Model"

_jcm, 2021, doi:10.3390/jcm10163585_

Round 1

Reviewer 1 Report

It is not clearly explained how the computer program works and how sensitivity and specificity are obtained with respect to ultrasound B-mode, making it difficult to interpret the results.
Guideline classification of polyps should be used (e.g. difference between true polyps and pseudopolyps). Conclusions too little developed.
This article could be interesting but I think that a publication like the one you propose need more elements and analysis to be of value for the Journal of Clinical Medicine. I think these criteria are not met by your manuscript.
Thank you very much for your effort and your submission.
Kind regards

Author Response

Response to Reviewer 1 Comments

Point 1: It is not clearly explained how the computer program works and how sensitivity and specificity are obtained with respect to ultrasound B-mode, making it difficult to interpret the results.

Response 1: Thank you for your comment. We agree that the overall diagnostic process and assessment of diagnostic performance have not been sufficiently described. A summary of the overall process is as follows.

We first established a gold standard for diagnosing GB polyps for training artificial intelligence ​​models and evaluating the diagnostic performance of artificial intelligence ​​models. This gold standard was the pathological diagnosis of GB polyps obtained from the cholecystectomy pathology report. The collected ultrasonography images were preprocessed for analysis. Then, we trained and validated the artificial intelligence ​​model by dividing the entire dataset into training and validation datasets. At this time, a five-fold cross-validation was performed to minimize the influence of how the training and validation datasets were divided and to obtain consistent results. Subsequently, the entire dataset was randomly divided into five equal sub-datasets. By changing the validation set each time, four of the five sub-datasets were used for training, and one was used for validation. In the training and validation processes, the probability that the patient’s GB polyp was a true polyp was calculated through the average value of the true polyp probability determined by the artificial intelligence ​​model for each of the patient’s multiple preprocessed ultrasound images. The final diagnosis of the artificial intelligence ​​model for each patient’s polyps was determined using the threshold that makes the Youden index the maximum value in each fold. Further, diagnostic performance measures, including the sensitivity and specificity of the artificial intelligence ​​model, were evaluated using the mean and 95% confidence interval of each fold outcome.

We have added this content in the “Materials and Methods” section (revised manuscript lines 78–81, 90, 156–164).

Point 2: Guideline classification of polyps should be used (e.g. difference between true polyps and pseudopolyps).

Response 2: Thank you for your comment. We agreement with your comment; thus, we have classified gallbladder polyps as a guideline classification of polyps (i.e., true polyps and pseudopolyps). As neoplastic polyps (including adenomas and adenocarcinomas that we classified) correspond to true polyps, and non-neoplastic polyps (including cholesterol polyps) correspond to pseudopolyps, we changed the terminology without changing the data. However, as many studies use the terms neoplastic and non-neoplastic in previous GB polyp research, we maintained some neoplastic/non-neoplastic terms in content related to those studies. Changes related to this content span the entire scope of the manuscript; hence, the revised manuscript lines are not displayed separately, but are highlighted in the revised manuscript.

Point 3: Conclusions too little developed.

Response 3: Thank you for your comment. We agree with your comment. The benefit that can be obtained in actual clinical practice through the current research results is quite limited. Further, we believe that it is necessary to develop a model that learn more data and a model using real-time ultrasound images and endoscopic ultrasound in the future. Therefore, we are planning follow-up studies and have added relevant limitations to the discussion section and directions for future research in the Conclusion section. We also added one reference accordingly. (revised manuscript lines 308–314, 319–321, 399–400)

Reviewer 2 Report

This paper evaluates in more detail about the interpretation of gallbladder polyps' malignancy by their ensemble convolutional neural network . Compared with the previous papers, limiting the size of polyps and excluding of the obvious malignant lesions contribute to more accurately evaluation.

It's trivial, but you don't have to have the paragraphs on lines 72-76.Regarding Fig. 2, it is easier to understand if there is a number of dataset cases. You should also list the number of cases used in the final diagnostic test.
It will be easier to understand if there are details about the ultrasonic device and the inspector, and ROI settings for how to create images to be trained.

I would appreciate it if you could consider using EUS and this Neural network model together in the future.  

Round 2

Reviewer 1 Report

Despite the effort of the authors to change the manuscript I think that this article cannot be an added value for the publication on the JCM.

Reviewer 2 Report

The authors have fulfilled each of the major compulsory revisions and modified the manuscript as requested. I have no further comment.